# The First National Survey of Cadmium in Cacao Farm Soil in Colombia

Daniel Bravo [1,*], Clara Leon-Moreno [1], Carlos Alberto Martínez [1], Viviana Marcela Varón-Ramírez [1], Gustavo Alfonso Araujo-Carrillo [1], Ruy Vargas [1], Ruth Quiroga-Mateus [1], Annie Zamora [2] and Edwin Antonio Gutiérrez Rodríguez [2]

1   Corporación Colombiana de Investigación Agropecuaria AGROSAVIA, C.I. Tibaitatá Km 14, Bogotá CO-0571, Colombia; cleon@agrosavia.co (C.L.-M.); cmartinez@agrosavia.co (C.A.M.); vvaron@agrosavia.co (V.M.V.-R.); garaujo@agrosavia.co (G.A.A.-C.); rvargas@agrosavia.co (R.V.); ryquiroga@agrosavia.co (R.Q.-M.)
2   Grupo de Investigación e Innovación en Cacao FEDECACAO, Federación Nacional de Cacaoteros–Fondo Nacional del Cacao, Bogotá, Colombia; annie.zamora@fedecacao.com.co (A.Z.); egragronomia@outlook.com (E.A.G.R.)
*   Correspondence: dbravo@agrosavia.co; Tel.: +57-(1)-422-7300 (ext. 1413)

**Abstract:** This study represents the first nationwide survey regarding the distribution of Cd content in cacao-growing soils in Colombia. The soil Cd distribution was analyzed using a cold/hotspots model. Moreover, both descriptive and predictive analytical tools were used to assess the key factors regulating the Cd concentration, considering Cd content and eight soil variables in the cacao systems. A critical discussion was performed in four main cacao-growing districts. Our results suggest that the performance of a model using all the variables will always be superior to the one using Zn alone. The analyzed variables featured an appropriate predictive performance, nonetheless, that performance has to be improved to develop a prediction method that might be used nationwide. Results from the fitted graphical models showed that the largest associations (as measured by the partial correlation coefficients) were those between Cd and Zn. Ca had the second-largest partial correlation with Cd and its predictive performance ranked second. Interestingly, it was found that there was a high variability in the factors correlated with Cd in cacao growing soils at a national level. Therefore, this study constitutes a baseline for the forthcoming studies in the country and should be reinforced with an analysis of cadmium content in cacao beans.

**Keywords:** agricultural soils; cacao plantations; soil cadmium content; soil parameters; spatial distribution

## 1. Introduction

Cadmium (Cd) is a nonessential heavy metal, which can be found in soil under many uses, including agriculture, pasture, forest and wasteland [1,2]. The presence of this metal in cacao-growing soils has become one of the biggest challenges to produce safe cocoa in South and Central American cocoa-producing countries [3–5]. It is one of the biggest challenges, because Cd is considered to be a primary soil pollutant, and is considered a major issue regarding human health, as high Cd concentrations have toxic effects on soil organisms, might easily transfer into the vegetation and ultimately enter the food chain [6]. Moreover, the presence of Cd soils in the three countries cited here (Colombia, Ecuador and Peru), should be addressed in two ways. One is from an agricultural perspective, the other, is a commercialization perspective due to the restrictions placed on Cd levels by the European Union commission [7]. On the one hand, regarding the soils where cacao is cultivated, the variability of Cd concentration in cacao beans from different sites has been attributed to the "total" soil Cd content and its relationship with critical soil factors influencing Cd phytoavailability, such as Cd soil mobility, pH, texture and soil organic matter (SOM) [8]. On the other hand, looking at the regulation of Cd levels in chocolate and cacao-derived

products imposed in 2019 [9], the negative impact of this to small scale producers during the commercialization should be a matter of new research. Henceforth, both geological and anthropogenic factors influence the presence of Cd in some cacao growing soils in Colombia. Therefore, assessing the cadmium distribution in cacao-growing soils is one of the first stages in developing strategies to address the issue. Cd is found in both the topsoil and the subsoils, interacting with the pedology, microbiology, and the weather [10]. This metal is associated with inorganic colloids, hydrated cations, in a free ionic form, or in complexes with organic or inorganic ligands [11]. Cd content in soils and its relationship with the pedosphere can vary depending on physical, chemical, and biological factors [12–14].

Since the presence of high levels of Cd in cacao is a regional issue (and not just in Colombia), several countries from Central and South America have been publishing data on the general geographical distribution of Cd both in soils and in cacao beans. For instance, in Ecuador, which produce large volumes of cacao [3], and Honduras, which produces small volumes, [5]. Moreover, Peru has begun studies of the spatial distribution of Cd content in both cacao-growing soil and cacao beans (Atkinson, Alliance Bioversity/CIAT International; personal communication) [15]. Colombia, which produces less cacao than Peru, has increased its interest in promoting cacao production throughout the country for internal use and export, as part of the government's plan to begin planting cacao in place of coca. However, to promote cacao, the distribution of soil Cd content is a critical criterion to the establishment of new plantations [4], and it might be helpful assess the beans' Cd content. Therefore, a spatial autocorrelation analysis of Cd content in the soil is an important tool for exploring the distribution and the spatial dependence on geographical phenomena [16].

It has been shown that among the related factors there are key parameters such as the soil elemental composition and soil pH levels that could influence Cd availability and accumulation in cacao-growing soils. Regarding the Cd content in the soil, it has been suggested that chemical parameters such as elemental cation exchange and the presence/competition of ions such as zinc (Zn) and manganese (Mn) between both the soil solution and the solid-state phase Cd-like compounds could influence Cd uptake by the cacao tree [4]. Soil pH has been shown to be the parameter with the most influence on the chemical composition of the pedosphere because of the geochemistry in both the topsoil and subsoils [17]. This is because of the ionic exchange capacity of Cd and Zn, which are very labile, so they react chemically with organic solutes, mineral silicates, and oxides in the microscale niche of the surrounding soil [18].

Previous studies have determined that the increase in Cd content in the soil causes changes in the absorption and accumulation ratios of macro- and micronutrients by cacao plants [19]. Several parameters may increase the Cd content in the soil. From these we highlight the composition and activity of microbial communities, where the absence of cadmium-tolerant bacteria (CdtB) populations might increase soil Cd content [10], a decrease in pH, a change in the redox potential, and the exuding of ligands. This is the first stage of metal ion uptake in plants to solubilize metals in the soil for their absorption [20]. Moreover, macronutrients such as calcium (Ca), potassium (K), and magnesium (Mg) are reduced in roots and stems in concentrations of 0.05 and 0.1 g.kg$^{-1}$ because of the Cd content in the soil [21]. Interestingly, the absorption of Ca by cacao roots has been recognized as a protective mechanism against Cd toxicity [22]. However, it is necessary to prove the relationship between all these parameters and Cd soil distribution, hence, the use of analytical tools such as neural networks [23], could improve our capacity to assess the key factors regulating the Cd fluxes.

Moreover, there has not been performed a national survey showing the distribution of Cd in cacao-growing soils in Colombia. Thus, this study constitutes the very first spatial distribution of Cd in cacao-growing soils using both Cd range scales and spatial correlation used to identify hotspots and coldspots. The present study aims to analyze the spatial distribution of Cd and the relationships between this heavy metal and the soil parameters in cacao-producing districts in Colombia, focusing on regions with the highest production

of cacao. The relationship is highlighted between the distribution of Cd in the soil and the soil parameters such as pH and the content of Ca, Iron (Fe), K, Mg, Mn, Phosphorus (P), and Zn. It also integrates a geographical analysis to identify polluted and partially polluted Cd cacao zones. Both descriptive and predictive statistical tools were used to assess the soil Cd distribution.

## 2. Materials and Methods

### 2.1. Sampling Design

The data used in this analysis were collected by the Federación Colombiana de Cacaoteros (FEDECACAO) and the Corporación Colombiana de Investigación Agropecuaria (AGROSAVIA, Mosquera, Colombia), between 2013 and 2016. The sample size and statistical distribution were determined according to the area planted with cacao in 19 districts in Colombia. For each of these districts, the number of farms to sample per municipality were based on the equation for finite populations [24]. The total points selected by the municipality represented 85% of the cacao growing area at the district level.

The soil samples were taken from cacao plantations with a minimum of three years since establishment and covering an area ≤5 ha (this is the average farm size for cocoa producers in Colombia). The soil samples were collected according to a previously reported method [24,25]. Zig-zag soil sampling was conducted, taking into account the uniformity, representativeness, and randomness of the plantation. The soil samples were collected around the trees, near the rhizosphere (points at least 70 cm from the tree trunk, on average), in the rhizosphere with the highest root density [26]. For each sample, both the soil litter and the very first surface soil boundary '*Ap*' were removed, assuring that the composite soil samples belong to the topsoil (30 cm soil in deep). The composite sample for each farm was obtained by homogenizing or mixing in a plastic recipient and taking out 500 g of soil that was stored in zip-bags. The sampling points were georeferenced for each farm in this survey.

### 2.2. Soil Type Description

Colombia has soil surveys on a general scale (1:100.000), carried out by the Instituto Geográfico Agustín Codazzi (IGAC). The soil surveys [27] show a series of delineated polygons based on qualitative soil characteristics e.g., the epipedon and endopedon diagnostic horizons and soil moisture regimes, and named cartographic soil units (CSU). Within the characteristics of the soil surveys of Colombia (IGAC, 2015), eight soil orders were identified in the soil samples, according to the USDA soil classification [27]. The orders belong to Inceptisols (63.1%), Entisols (20.8%), Andisols (5.0%), Mollisols (4.9%), Oxisols (4.0%), Ultisols (1.3%), Alfisols (0.7%) and Vertisols (0.2%).

From these, Inceptisols are mainly found in the Santander district. Inceptisols have a predominant udic soil moisture regime. These soils have been developed from sedimentary and metamorphic rocks with coarse, medium, fine, and very fine grain size. Moreover, Inceptisols are in all topographic slopes. Entisols are in the Santander and Arauca districts. Entisols in Arauca are predominant in alluvial soils in the valley landscape formed by the Arauca river, these soils have been formed from sand deposits and they have a predominant udic soil moisture. Likewise, Andisols were described for Antioquia, Tolima, and Nariño; these soils are in a mountain landscape formed by volcanic activity. Mollisols are found in Huila, Santander, Risaralda and Valle del Cauca districts, these soils have udic soil moisture and, in some places, vertic subgroups.

### 2.3. Chemical Analysis

The soil samples were pretreated by a drying process with a temperature of 40 °C and sieved using a 2 mm wide mesh. The pseudototal Cd content in the soil was determined according to previous studies [10,28], based on spectrometric measurements. The digestion was carried out in 1 g of soil, using nitric acid (HNO$_3$)—perchloric acid (HClO$_4$), 9:3 *v/v* and a heating plate. Cd concentration was determined by inductive coupled plasma spectrometry with optical emission—ICP-OES (Thermo Scientific ICAP 6000, Waltham, MA,

USA). It is a pseudototal, because it represents the method recommended by the Environmental Protection Agency (EPA 3050B) which was used as the conventional pseudototal digestion method [29], and digests and extracts cadmium using strong acids such as, HCl, HNO₃, HClO₄, but not less hydrofluoric acid, so that only the heavy metals retained in the more labile soil fractions such as organic matter or carbonates are released into the extractant solution [30]. Moreover, pseudototal soil cadmium concentrations have been significantly ($p < 0.001$) more highly correlated with bean cadmium in sites at higher elevations and in a temperate, drier climate, and cacao cultivars in farms assessed in Peru and Colombia [4,31]. The efficiency of the processes was evaluated in terms of the percentage of recovery of the reference material for soils (ISE sample 961—Wepal, Wageningen, The Netherlands), determined through the relationship between the pseudototal Cd in the extracts of the soil samples and the pseudototal concentration present in the soil reference material, after being exposed to acid digestion. The determination of Cd concentration was performed with calibration curves measured with ICP—OES. A reference material of pure cadmium (Merck—SRM traceable standard solution of NIST Cd $(NO_3)_2$ in $HNO_3$ 0.5 mol/L—1000 mg.kg$^{-1}$ Cd Certipur. Reference 1.19777.0500) was used to establish the standard calibration curves. Low, medium, and high range calibration curves were prepared, with 10 points each, where the detection limit of quantification was 0.040 mg.kg$^{-1}$ of $Cd^{2+}$, and Cd recovered in soil samples was 99.3%. The available micronutrient contents, Fe, Mn, and Zn were determined by the Mehlich method [32]. The quantification was done using atomic absorption spectrometry equipment -AAS (Agilent 280FS AA, Santa Clara, CA, USA). In addition, Ca, K, and Mg content were determined using the method published elsewhere [33]. The soil pHH₂O was quantified using a potentiometer, with pH Electrode InLab Max Pro (Automatic Titrator T90, Mettler Toledo, Columbus, OH, USA). The pH values were obtained in a soil:water solution (1:2.5 $w/v$). The content of P was determined using the Bray II extraction method [34]. The P content was measured using a UV−Visible Spectrophotometer (Perkin Elmer spectrometer Lambda 25, Waltham, MA, USA) at 887 nm.

### 2.4. Database Setting and Editing

Once AGROSAVIA and FEDECACAO datasets were merged, the data were edited and verified. Observations that did not meet the expected requirements were discarded. Hitherto, Cd values outside the interval (0.01 mg kg$^{-1}$, 30 mg kg$^{-1}$) were discarded. Similarly, Fe, Mn, Zn, Ca, K, Mg, and P were screened to detect atypical values. Finally, samples with at least one missing record for these variables were removed.

### 2.5. Data Analysis

#### 2.5.1. Descriptive Analysis

For all soil properties studied, the samples underwent descriptive and exploratory analyses at a district level. Measures of central tendency and dispersion were calculated [35]. For atypical data, each soil property was reviewed to identify soil observations outside the ranges of variables. Then, a correlogram with the 8 soil parameters and the Cd content of the soil was developed.

#### 2.5.2. Ability of Soil Parameters to Predict Cd Content

The aim of this analysis was to estimate the collective and individual predictive capability of pH, Ca, Fe, K, Mg, Mn, P, and Zn (hereafter called 8P). Two error/loss functions were considered (mean squared error (MSE) and mean absolute error (MAE)), and the generalization error was estimated using five-fold cross-validation [36]. Moreover, the cross-validation (CV) estimates of Pearson's correlation coefficient between observed and predicted Cd values (PC) also were computed. Several methods/models and basis expansions of predictive variables were considered. Their combinations yielded different predictive machines, and those with smaller MSE and MAE and larger PC values were preferred.

Five approaches were used to build the predictive machines: multiple linear regression using ordinary least squares (MLR-OLS), the least absolute shrinkage and selection operator (LASSO) or ridge regression (MLR-RR) to estimate model location parameters, as well as artificial neural networks (ANN) considering different architectures, and random forest (RF). Moreover, the following sets of input variables were considered: set 8P (LEO), 8P plus their corresponding quadratic expansions (LQE), and 8P plus their corresponding exponential expansions (LEE). The spatial location of the cacao farms (LL) also was considered as explanatory variable. Thus, for each one of these sets, there were two machines, one with LL and another without LL. In addition, some categorical variables were considered by precorrecting Cd using additive correction factors. The factors were laboratory, sample depth, and agroecological zone. The penalty parameters for MLR-LASSO and MLR-RR were tuned implementing a five-fold CV scheme.

### 2.5.3. Estimation of Association Networks

Partial correlation networks are a useful tool to elucidate complex patterns of association between sets of random variables. So, a concentration graph model was used to estimate the partial correlations between nine variables: Cd and those in 8P. Preliminary analyses using Mardia and Hene−Zikler tests, resulted in rejection of the null hypothesis of multivariate normality ($p < 0.05$). Therefore, the Convex Correlation Selection Method-CONCORD [37], which does not depend on that assumption, was used to estimate the partial correlation network, which describes association patterns between pairs of variables after controlling for the remaining ones. The BIC-like criterion previously described [37] was used to set CONCORD's tuning parameter. Data editing, as well as predictive and partial correlation network analyses were carried out using R version 4.0.1 [38].

### 2.6. Cd Distribution Assessment in Cacao-Growing Soils

#### 2.6.1. Distribution of Hotspots and Coldspots

The spatial point distribution of the Cd content of the soil was performed using the Getis-Ord ($G\_i\hat{~}^*$) spatial autocorrelation technique [39]. This method assesses the degree by which each sampled point is surrounded by other points with similar values, either higher or lower in comparison, within a specified geographical distance or threshold distance '$d$' [40].

The spatial autocorrelation analysis was implemented using the geographic information system software ArcGIS (Esri, Redlands, CA, USA). The index $G\_i\hat{~}^* > 0$ denotes a spatial dependence among high values (geographical hotspots), and $G\_i\hat{~}^* \leq 0$ refers to a spatial dependence for low values (geographical coldspots). Values close to or equal to zero indicate there was no significance [40]. A high value (Cd concentration) means that a point has high concentration but is surrounded by high values. A low value mean that a point has low concentration but is surrounded by low values. Therefore, high, or low values, here, denoted both Cd concentration and spatial dependence. The degree of clustering and its statistical significance was evaluated based on the outputs *z*-scores and *p*-values ($p < 0.10$, $p < 0.05$, and $p < 0.01$).

The software tool's "false discover rate" (FDR) was applied to reduce the critical *p*-value thresholds, to produce multiple tests and spatial dependency. The "$d$" depends on the use of the incremental spatial autocorrelation tool. This was an iterative and data-driven process that assessed the spatial autocorrelation variations at different distances.

#### 2.6.2. Analyzing the Main Cacao-Producing Districts by Cold/Hotspots

Four out of 19 cacao-growing districts in Colombia were selected to analyze the relationship between the 8P and the Cd content of the soil. The districts were Antioquia, Arauca, Huila, and Santander. Together they produce more than 65% of Colombia's cacao [25]. Because of its extensive cacao tradition, Santander was the district that produced the most cacao (25,158 tons or 42.1%) of the national production. It was followed by Antioquia, where the production ratio rose from 3553 tons per ha in 2014 to 5259 tons per ha in 2019 (8.82%). Arauca also has been a great cacao-producing district, at 4546 tons (7.6%), and the Huila district produced 4051 (6.8%) of the national production [25].

## 3. Results

### 3.1. Descriptive Analysis of the Soil Samples

Figure 1 shows the geographic distribution of Cd in the soil of the samples in the cacao production areas. The color scale shows the levels of Cd concentration. The districts of Santander and Boyacá showed the highest Cd values (red dots). Antioquia, Caquetá, Guaviare, Huila, Meta, Risaralda, and Tolima showed Cd concentrations $\leq 2$ mg kg$^{-1}$ (green and yellow dots).

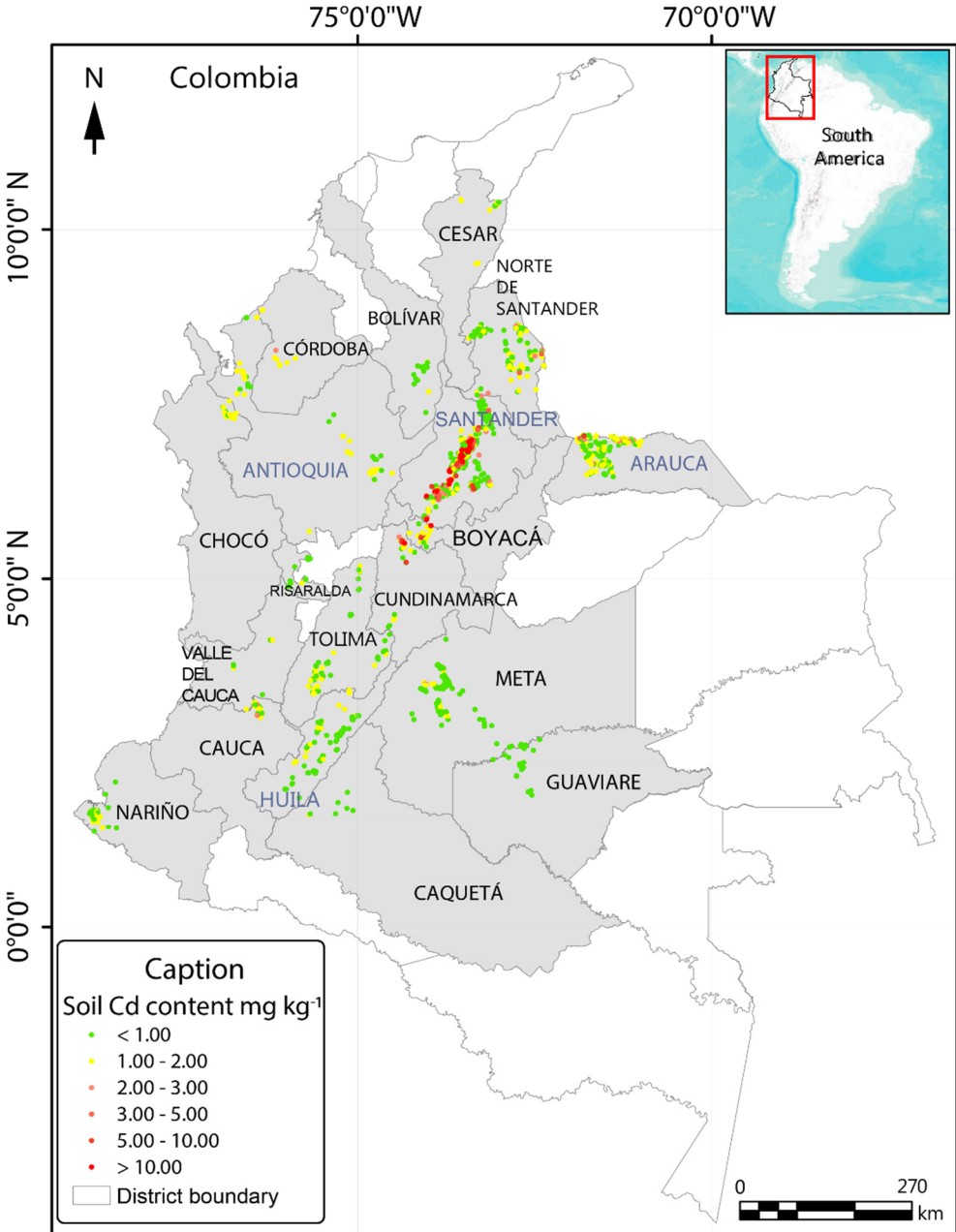

**Figure 1.** Distribution of pseudototal Cd in the soil of 1837 cacao-growing farms in Colombia. Grey districts correspond to the existing data. White districts belong to areas where no pseudototal Cd data was found.

Table 1 shows the statistics for concentrations of Cd in the soil samples at the district level. Cadmium concentration in the soil samples ranged from means of 0.01 mg kg$^{-1}$ to 27 mg kg$^{-1}$. Regarding Cd by cacao-growing districts, the soil samples in Santander ($n = 821$) had an average Cd content in the soil of 1.9 mg kg$^{-1}$, above the threshold value for agricultural soils [41], next were Boyacá (1.82 mg kg$^{-1}$, $n = 17$) and Córdoba (1.63 mg kg$^{-1}$,

$n = 11$). Lower values were found in Bolívar (0.4 mg kg$^{-1}$, $n = 17$), Guaviare (0.44 mg kg$^{-1}$, $n = 25$) and Caquetá (0.45 mg kg$^{-1}$, $n = 7$). The Santander district showed the highest variability (Cv 1.73), followed by Bolívar (Cv 1.26), Cundinamarca (Cv 1.01), and Boyacá (Cv 0.86). The lowest Cv's were observed in Córdoba (Cv 0.19), Chocó (Cv 0.22), and Valle del Cauca (Cv 0.22). Therefore, the spatial variability was not higher at the national scale (0.12 < Cv < 0.6). As shown in Supplementary Figure S1, the Cd concentration had a linear positive correlation with Zn, Ca, P, and pH (with correlation coefficients of 0.61, 0.51, 0.33, and 0.31, respectively).

**Table 1.** Concentrations of Cd (mg kg$^{-1}$) in the soil assessed in cacao-growing districts of Colombia.

| District | $n$ | Mean | Median | Min | Max | SD | Cv | Sk | Kt |
|---|---|---|---|---|---|---|---|---|---|
| Antioquia | 60 | 1.14 | 1.19 | 0.20 | 1.86 | 0.39 | 0.34 | −0.46 | −0.39 |
| Arauca | 232 | 1.11 | 1.05 | 0.28 | 5.64 | 0.61 | 0.55 | 2.33 | 12.80 |
| Bolívar | 17 | 0.40 | 0.01 | 0.01 | 1.41 | 0.51 | 1.26 | 0.58 | −1.42 |
| Boyacá | 17 | 1.82 | 1.25 | 0.56 | 6.17 | 1.62 | 0.89 | 1.90 | 2.21 |
| Caquetá | 7 | 0.45 | 0.41 | 0.04 | 1.19 | 0.36 | 0.81 | 0.95 | −0.30 |
| Cauca | 47 | 1.32 | 1.33 | 0.45 | 2.34 | 0.38 | 0.29 | 0.20 | 0.06 |
| Cesar | 20 | 1.01 | 1.09 | 0.31 | 1.63 | 0.38 | 0.38 | −0.53 | −0.71 |
| Chocó | 27 | 1.31 | 1.24 | 0.85 | 1.91 | 0.29 | 0.22 | 0.53 | −0.56 |
| Córdoba | 11 | 1.63 | 1.68 | 1.11 | 2.04 | 0.31 | 0.19 | −0.21 | −1.44 |
| Cundinamarca | 32 | 2.83 | 1.60 | 0.53 | 13.30 | 2.86 | 1.01 | 1.81 | 3.29 |
| Guaviare | 25 | 0.44 | 0.36 | 0.14 | 0.99 | 0.23 | 0.53 | 0.91 | −0.05 |
| Huila | 105 | 0.83 | 0.83 | 0.29 | 1.49 | 0.26 | 0.31 | 0.33 | −0.08 |
| Meta | 116 | 0.61 | 0.65 | 0.01 | 2.29 | 0.43 | 0.70 | 0.40 | 0.71 |
| Nariño | 45 | 0.91 | 0.93 | 0.20 | 1.38 | 0.24 | 0.27 | −0.64 | 0.82 |
| Norte de Santander | 150 | 1.08 | 0.97 | 0.09 | 4.51 | 0.62 | 0.58 | 2.06 | 7.01 |
| Risaralda | 12 | 0.70 | 0.74 | 0.21 | 1.01 | 0.22 | 0.31 | −0.65 | −0.38 |
| Santander | 821 | 1.90 | 0.83 | 0.01 | 27.00 | 3.29 | 1.73 | 3.87 | 17.64 |
| Tolima | 84 | 0.98 | 0.91 | 0.24 | 1.77 | 0.34 | 0.35 | 0.34 | −0.43 |
| Valle del Cauca | 9 | 0.91 | 0.86 | 0.63 | 1.16 | 0.20 | 0.22 | −0.08 | −1.75 |

$n$: number of the soil samples, SD: standard deviation, Cv: coefficient of variation, Sk: coefficient of skewness, Kt: coefficient of kurtosis.

### 3.2. Ability of Soil Parameters to Predict Cd Content

Across all combinations of input variables, response variables, and statistical models/methods, there were no marked differences in predictive performance. RF had a superior predictive capability; it outperformed the other methods in all settings except one (LEO as inputs and Cd as response variable). Under this setting, an ANN with a single hidden layer and a hidden unit achieved the best performance, as measured by MSE, MAE, and PC (Table 2). However, RF yielded very similar results. Although the ANN models considered in this study had simple architectures, most of them exhibited convergence problems. Even though ANN yielded the best results under the stated conditions, the general performance of these models was poor (Supplementary Table S1). Moreover, predictive machines based on multiple linear regression models performed similarly across all settings. Table 2 summarizes the results of the predictive analyses. For some combinations of input-response variables, it shows the model/method with the best performance along with the corresponding MSE, MAE, and PC values. Results for Cd pre-corrected for sample depth or agroecological zone are not shown because their performance was inferior.

The predictive machine with the best overall performance was an RF with LEO plus LL as explanatory variables and Cd precorrected for laboratory effect as the response variable. When assessing the individual predictive performance of variables in set 8P, Zn, P, and Ca yielded the best performance. Supplementary Table S1, shows CP, MAE, and MSE values obtained using the best approach for each variable.

**Table 2.** Results of the predictive analysis.

| Input Variables | Response Variable * | Model/Method | PC | MSE | MAE |
|---|---|---|---|---|---|
| LEO | Cd | ANN | 0.73 | 2.51 | 0.72 |
| LEO + LL | Cd | RF | 0.73 | 2.50 | 0.68 |
| LQE | Cd | RF | 0.71 | 2.63 | 0.73 |
| LQE + LL | Cd | RF | 0.72 | 2.58 | 0.70 |
| LEE | Cd | RF | 0.72 | 2.61 | 0.73 |
| LEE + LL | Cd | RF | 0.72 | 2.55 | 0.70 |
| LEO | Cd.PCorr.Lab * | RF | 0.73 | 2.59 | 0.73 |
| LEO + LL | Cd.PCorr.Lab | RF | 0.74 | 2.49 | 0.68 |
| LQE | Cd.PCorr.Lab | RF | 0.72 | 2.62 | 0.73 |
| LQE + LL | Cd.PCorr.Lab | RF | 0.72 | 2.65 | 0.70 |
| LEE | Cd.PCorr.Lab | RF | 0.72 | 2.60 | 0.73 |
| LEE + LL | Cd.PCorr.Lab | RF | 0.72 | 2.65 | 0.70 |

* Cd.PCorr.Lab: Cd precorrected for laboratory effects. Cd: cadmium, LEO: set of 8 soil variables, LL: spatial location of the cacao farms, LQE: set of 8 soil variables and their corresponding quadratic expansions, LEE: set of 8 soil variables and their corresponding exponential expansions, ANN: artificial neural networks, RF: random forest, PC: predictive correlation, MSE: mean squared error, MAE: mean absolute error.

Estimation of Association Networks

Estimated partial correlation networks were very similar when using Cd precorrected for laboratory effect or Cd without precorrection, so, the analysis based on Cd is discussed (see Supplementary Tables S2 and S3). According to these results, Ca, P, and Fe had the highest estimated degrees, while K and Mg had the lowest. Regarding estimated partial correlations, those between Cd and Zn (0.34), Ca and pH (0.22), and Ca and Cd (0.18) were the largest (Supplementary Table S2). The estimated network density (the ratio of the number of non-null partial correlations to $0.5 \times p \times (p - 1)$, the number of partial correlations) was 0.6389. Table S3, shows the estimated degrees (number of connections to other nodes) of each variable. Figure 2 shows the inferred partial correlation network.

### 3.3. Analyzing the Cd Distribution by Hotspot and Coldspot

The optimal fixed distance band was 42 km. This distance ensured that all features had at least one neighbor. The spatial structure of the data had (i) an average number of neighbors of 188, (ii) a minimum number of neighbors of 1, and (iii) a maximum number of neighbors of 519. Soil samples from farms with fewer neighbors were observed in the districts of Antioquia, Bolívar, Cesar, Meta, Nariño, Tolima, and Valle del Cauca (Figure 3). Statistically significant were 1055 output features based on an FDR correction for multiple testing and spatial dependence (Table 3).

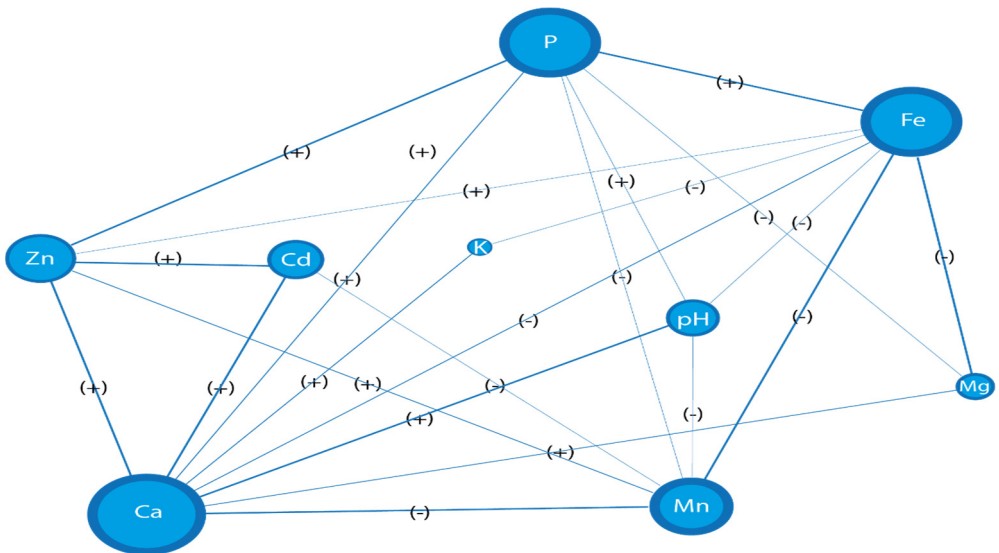

**Figure 2.** Inferred partial correlation network. Edge widths are proportional to the magnitude of the corresponding partial correlation coefficient whereas node sizes are proportional to the corresponding variable degree.

**Table 3.** Results of the hotspot and coldspot analysis both nationwide and in four main cacao-growing districts.

| Place | Type | Number | Confidence Level [%] | Number |
|---|---|---|---|---|
| National | Hotspot | 703 | 99 | 698 |
| | | | 95 | 2 |
| | | | 90 | 3 |
| | Coldspot | 352 | 99 | 168 |
| | | | 95 | 68 |
| | | | 90 | 116 |
| | Randomless | 782 | Not significant | 782 |
| Santander | Hotspot | 660 | 99 | 657 |
| | | | 95 | 1 |
| | | | 90 | 2 |
| | Coldspot | 89 | 99 | 88 |
| | | | 90 | 1 |
| | Randomless | 72 | Not significant | 72 |
| Arauca | Coldspot | 108 | 95 | 45 |
| | | | 90 | 63 |
| | Randomless | 124 | Not significant | 124 |
| Huila | Coldspot | 105 | 95 | 12 |
| | | | 90 | 20 |
| | Randomless | 73 | Not significant | 73 |
| Antioquia | Randomless | 60 | Not significant | 60 |

*3.4. Pseudototal Cd Content in the Productive Cacao-Growing Districts in Colombia*

Figure 4 shows the distribution of Cd in soils from 4 out of 19 cacao-growing districts. Of all the districts analyzed here, Santander had the widest range of Cd concentrations showed in Figure 4B, (ranging 0.01–27 mg kg$^{-1}$), compared to Arauca in Figure 4D (ranging 0.28–5.64 mg kg$^{-1}$), Huila in Figure 4C (ranging 0.29–1.49 mg kg$^{-1}$) and Antioquia in Figure 4A (ranging 0.2–1.86 mg kg$^{-1}$).

These results are analyzed in more detail in Section 4. It is important to note that a major sampling effort was made in Santander (with 821 assessed farms) compared to Arauca (232 farms), Huila (105 farms), and Antioquia (60 farms). The reference threshold used in panels A−D in Figure 5, corresponds to the maximum allowable Cd content of 0.8 mg kg$^{-1}$ in chocolate that has ≥50% of cocoa solids in the final product, according to the European Union regulation [7,9] just to compare in a possible relationship between the soils' and the beans' Cd content.

Figure 5 shows the distribution of Cd content in the four main cacao-growing districts in Colombia. The highest variability in Cd content occurred in cacao farms from Santander (see Figure 5D), whereas the least variability was observed in the Antioquia district (Figure 5A). Meanwhile, both Arauca and Huila (Figures 5B,C, respectively), have shown medium variability regardless of the sample number. It should be noted that, even if Santander showed high variability and had the highest Cd content, the concentration of farms with a <1 mg kg$^{-1}$ of Cd content in the soil was widely distributed across the district. Interestingly, in the Arauca district, samples with >10 mg kg$^{-1}$ of Cd content in the soil were all located in the basin of the Arauca river or near the slopes of this river (See Figure 5B).

The main Cd content in the soil by district is shown in Table 1. The Santander district proved to have the highest numbers of both soil samples (821) and hotspots (660). The mean value and the standard deviation (SD, represented by the symbol ±) of Cd in Santander were 2.18 ± 3.58 mg kg$^{-1}$ in hotspots, 0.51 ± 0.57 mg kg$^{-1}$ in coldspots, and 1.02 ± 1.26 mg kg$^{-1}$ in a nonsignificant range. In contrast, the other districts did not show hotspots, and in the Antioquia district, there were no coldspots either. In the Arauca district, the mean and SD Cd level were 0.94 ± 0.40 mg kg$^{-1}$ in coldspots and 1.26 ± 0.71 mg kg$^{-1}$ for nonsignificant values. In Huila, the Cd mean, and SD values were 0.75 ± 0.24 mg kg$^{-1}$ in coldspots and 0.86 ± 0.26 mg kg$^{-1}$ in the nonsignificant range, whereas in the Antioquia

district, the mean and SD Cd content in the soil were $1.14 \pm 0.39$ mg kg$^{-1}$. The SD is higher than the mean in several cases due to the high scatter of the data in the hotspot and coldspot analysis.

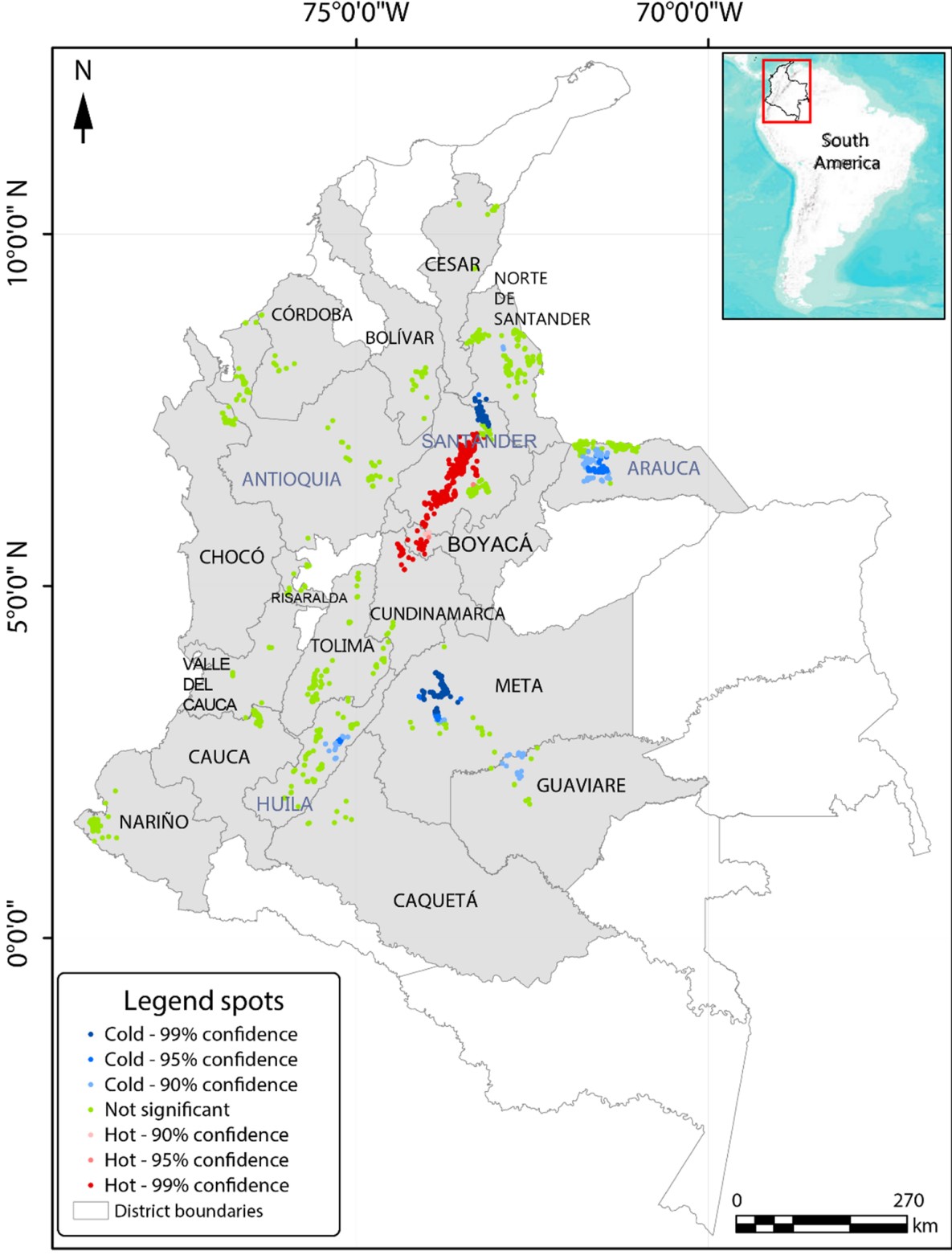

**Figure 3.** Analysis of hotspots (**red dots**) and coldspots (**blue dots**) of the studied area.

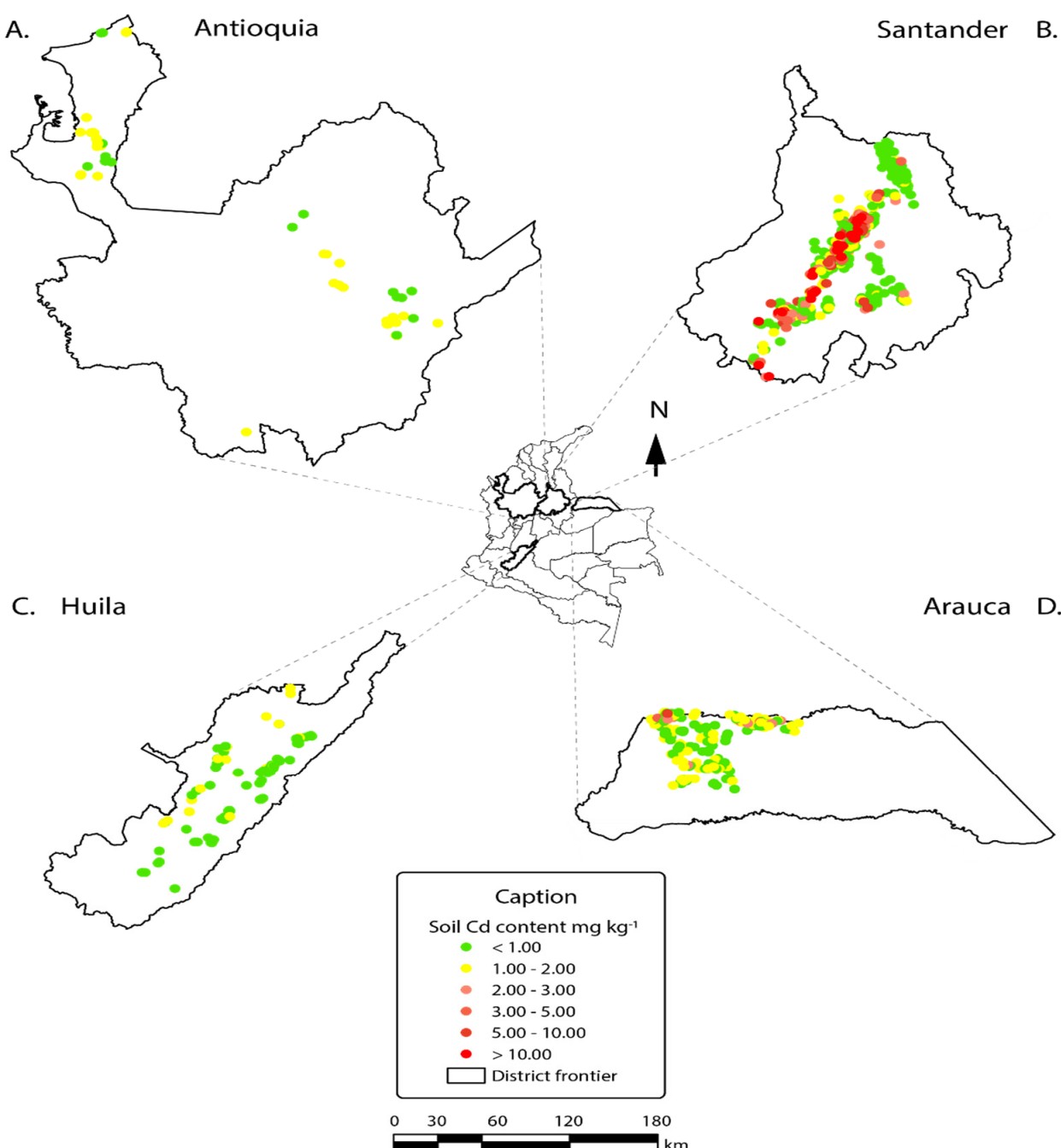

**Figure 4.** The relationship between cadmium content (Cd) in the soil and the number of farms assessed in four out of nineteen districts selected for comparison. It is highlighted the bigger number of samples in Santander, regarding other districts. (**A**). Antioquia district, (**B**). Santander district, (**C**). Huila district, (**D**). Arauca district.

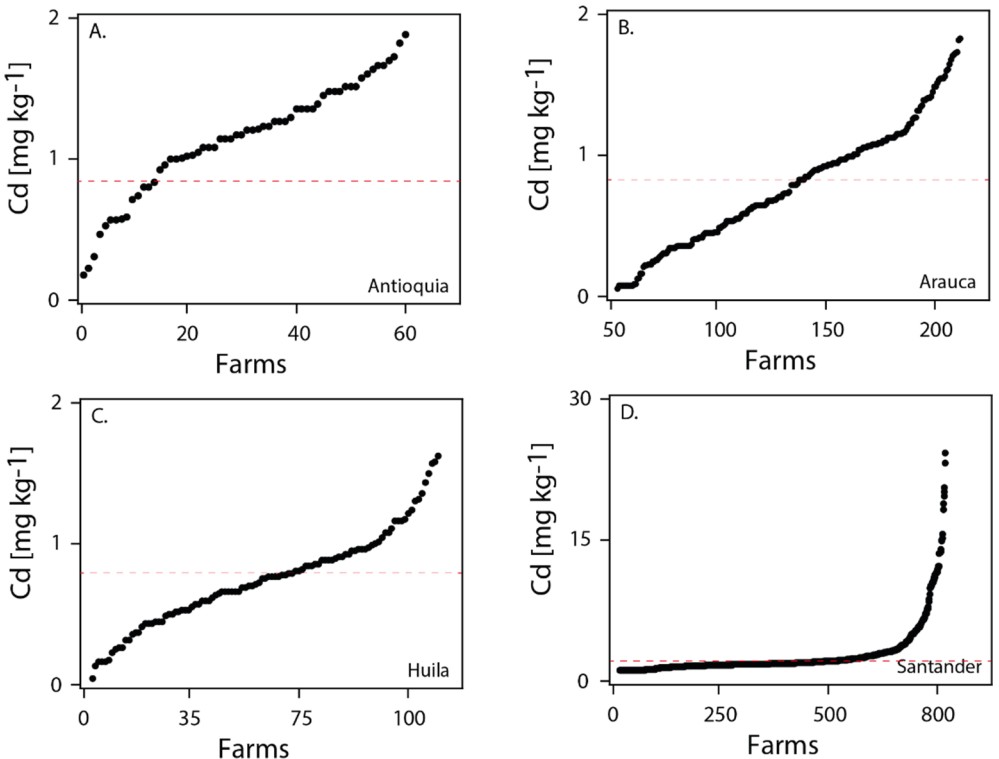

**Figure 5.** Distribution of cadmium (Cd) regarding four main cacao productive districts in Colombia. The districts correspond to (**A**) Antioquia, (**B**) Arauca, (**C**) Huila, and (**D**) Santander, showing the relative variability of Cd ranges found in each district.

## 4. Discussion

In this data set, 777 out of 1837 soil samples exhibited cadmium concentrations higher than the threshold value established for agricultural soils by the Ministry of the Environment of Finland [41] and used in the study of cacao-growing farms in Colombia [4]. There is substantial variation in the threshold values of heavy metal concentrations in soil. This variation is due to the environmental risk assessment models, ecotoxicological criteria and human toxicological parameter values [42]. The Finland Decree [41] has been applied in an international context for agricultural soils by UNEP [43] and at a continental level in a study of heavy metals in agricultural soils of the European Union [44]. Additionally, the same threshold value (1 mg kg$^{-1}$) for agricultural soils has been used in countries such as Sweden, Germany, Belgium and Austria [45]. Regionally, Peru uses the Canadian standard of 1.4 mg.kg$^{-1}$ (Atkinson, personal communication). Unfortunately, Colombia does not have a soil environmental quality standard yet.

The average soil Cd content in Colombia (1.43 mg kg$^{-1}$) was higher than the threshold value (1 mg kg$^{-1}$) established by different countries such as Finland, Sweden, Belgium, and Austria [43–45]. In the regional context, this average was higher than that reported in Ecuador, 0.44 mg kg$^{-1}$ [2], Bolivia, 0.3 mg kg$^{-1}$ [35], or Honduras, 0.25 mg kg$^{-1}$ at a soil depth of 0–10 cm and 0.16 mg kg$^{-1}$ at a soil depth of 10–25 cm [5]. Moreover, in non-cacao-growing soils, in Brazil, Cd concentrations between 0.3 and 1.6 mg kg$^{-1}$ have been reported [46]. In our study, the samples from the Cundinamarca district ($n = 32$) had the highest mean for Cd concentration (2.83 $\pm$ 2.86 mg kg$^{-1}$). These values were lower than previous studies [47] where soil Cd content was found to be 10.68 mg kg$^{-1}$ and 7.92 mg kg$^{-1}$ at soil depths of 0–30 cm and 60–100 cm, respectively.

With respect to the correlation of Cd content and soil properties, other studies have reported a linear correlation even in natural soils, such as in Brazil, with ratios of 0.77 and 0.86 for Cd/Zn and Cd/Ca, respectively [48]. In contrast, cacao-growing soils in Ecuador

had a stepwise multivariate correlation of 0.37 between Cd and pH, which include the SOM [3].

The spatial autocorrelation analysis of the Cd content in the soil showed three types of data: one set had high values, another had low values, and the third had samples that were in the middle. It is important to note that the spatial autocorrelation analysis was performed at the national level, not at the district level. We obtained a distance band equal to 42 km. That distance is very important in the calculation of spatial autocorrelation and may vary if the analysis is national or local. The set with high values was located mainly in the district of Santander. To understand the greater importance of hotspots found in the Santander district, the analysis should include more factors surrounding the pedology and climate of the cacao-growing farms assessed. Some studies assessed only the organic matter [19]; parental material [13]; geological and topographic characteristics [49], or as an input due to pollution from human sources in the agroecosystem [50].

The Cd counts in the sediment and surface river fluxes had been analyzed before [51]. This study highlights the Cd content of the surface sediments (collected on the banks), and it reports four hydrographic subzones that had high levels of Cd. These zones correspond to the Marmato river in the district of Caldas (center-western Colombia), the Bogotá river in the district of Cundinamarca (the middle of the country), the Negro river in Cundinamarca, and the Carare-Minero river within the districts of Boyacá and Santander. It is important to note that, in the national analysis, the last two areas mentioned included samples with hotspots. Specific studies have also been developed in Colombia to analyze the source of Cd from water resources in cacao plantations. For example, the Institution Centro de Productividad y Competitividad del Oriente—CPC [52] carried out a survey analyzing 124 water samples in the six municipalities with the highest concentration of Cd within the Santander district. They concluded that the water tributaries and wastewater did not significantly influence the high concentration of Cd in the soils of that area. Regarding the geochemical atlas of Colombia, which included Cd counts in subsoil [53], the hotspots spatially assessed in this study coincided with the areas reported in the atlas.

However, it is important to note that the methodologies, sampling techniques, and analytical parameters of the studies were quite different from each other, so the results are not comparable. Nonetheless, because the highest values coincided spatially, more-detailed research of the Santander district should be conducted. For example, in the municipality of San Vicente de Chucurí, located in Santander, the Umir formation, which is a lithostratigraphic unit of the upper cretaceous system, is widely distributed. That formation has a thickness between 800 m and 1400 m, made up of sedimentary rocks like lodolites and claystones and some bituminous coal [54]. Bituminous coal mantles have been evaluated in the Ranchería and Cesar basins, where soil cadmium content above the world averages in the Earth's crust were found [55].

Regarding the coldspots, the comparison of our findings with the geochemical atlas was incomplete because many of the areas assessed in this study were not reported in the atlas. Interestingly, in the areas where the studies overlap, especially in the north of Santander, the low concentrations reported in the atlas and in the current coldspot analysis coincided. An in-depth study should be conducted in Santander to identify the Cd dynamics in both low and high Cd content scenarios.

### 4.1. Ability of Soil Parameters to Predict Cd Content

Of the three factors considered for Cd precorrection, the laboratory had the best predictive performance. In a previous work [56], it was noted that "significant" variables (variables showing a strong association with the response variable) are not necessarily good predictors. Assessing the significance of these factors is beyond the scope of this study, so procedures to infer it were not carried out. However, although one could point to the laboratory setting, sample depth, or agroecological zone as "relevant" variables for predicting Cd, our results suggest that they were not, even though those factors could be "significantly" associated with Cd.

As to the predictive ability of individual variables in set 8P, regardless of the statistical method/model used to train the predictive machine, Zn always had the best performance which means that this parameter had a better predictive ability or predictive performance. However, our results suggest that the performance of a machine using all the variables in 8P will always be superior to using Zn alone. The predictive capability of the best performing machine found in this study could be considered satisfactory according to the ratios of MAE to Cd mean and median, 0.48 and 0.76, respectively. Similarly, the ratios of the positive square root of MSE to Cd mean and median were 1.10 and 1.75, respectively. Smaller ratios indicate better performance, and CP values closer to 1 are desirable.

It is worth mentioning that, since this is the first effort at Cd prediction in Colombia, this study was based on the available data, but there are more variables to be considered in future analyses and using them could improve predictive capability. Those variables include physical characteristics of the soil such as texture or bulk and particle densities, cacao varieties, plant sections in both the scion and rootstock, the age of the crop, the level of precipitation, the type of soil (geogenesis), and the characterization of the microbiota with the ability to immobilize Cd, a bacterial functional group called Cadmium-tolerant bacteria (CdtB) [10]. Furthermore, other statistical models/methods, such as deep neural networks, could be used to build the predictive machine.

Thus, we recommend enlarging the dataset on two levels: (i) the variables, and (ii) the number of observations, and then training several predictive machines before publishing a prediction equation. In summary, it can be said that the variables in set 8P featured an appropriate predictive performance, but that performance must be improved to develop a prediction method that can be used nationwide.

### 4.2. Estimation of Association Networks

Results from the fitted graphical models agreed with those from predictive analysis because the largest associations (as measured by the partial correlation coefficients) were those between Cd and Zn. As mentioned above, Zn was found to be the explanatory variable with the highest individual predictive performance. Similarly, Ca had the second-largest partial correlation with Cd and its predictive performance ranked second (Supplementary Table S4). Seeing the inferred network as a system, variable degrees have been used as indicators of relevance in that system, and this approach has proven to be useful in other studies [57]. Thus, using this criterion, our results pointed to Ca, P, and Fe as relevant variables in this system (Figure 2 and Supplementary Figure S1). The estimated network also revealed the importance of Zn and Ca to describe and predict Cd, not only because of their direct relation to this variable, which had some of the highest partial correlation coefficients, but also because of their indirect relationship through other pathways, as shown in Figure 2.

### 4.3. Biological Meaning of the Findings

A previous study was carried out regarding not only Cd, but also As, Hg and Pb in cacao growing soils from northeast Colombia [58]. The study showed that Cd was the metal of most concern regarding limits of heavy metals in chocolate at an international level.

Our findings in this survey show that both the geographical distribution of Cd and the number of cacao-growing farms assessed in each of the four districts, might exert an influence on the variability of Cd in cocoa beans, a relationship that should be confirmed. This relationship must include other types of study, for example, microbiological evaluations, eco-physiological description, genetic analysis, and absence or presence of other plant species that reduce the concentration of Cd [59].

Both Cd and Zn, at high concentrations, can cause plant toxicity [60] due to their physical and chemical similarities, considering that both belong to group II of the periodic table and are generally found associated ions, competing with each other for several binders [3,13,14]. It has been shown that the translocation efficiency of Cd and Zn is highly correlated, suggesting the possibility of a common transport mechanism for both metals, meaning that the accumulation of Cd in plants might be modulated by the presence

of Zn [60]. Regarding the relationship between Cd and Ca, geochemical indicators of cadmium bioweathering in cacao-growing systems are emerging as important predictors of soil Cd dynamics [61], nonetheless, evidence concerning the role of calcium (Ca) in this process is scarce. The bioweathering geochemical pathway mediated by cadmium-tolerant bacteria (CdtB), may release $CaCO_3$ from cacao sites due to the influence of Ca on aggregation, inhibiting oxidative transformation, and preserving/stabilizing lower soil organic carbon [61–63]. Since calcite ($CaCO_3$) and otavite ($CdCO_3$) are minerals slightly soluble in water, a Cd and Ca sink might exist in cacao-growing subsurface soils. Evidence of this has been found by analyzing with XRD colloids and mineral formations around cacao roots in situ [10].

Among the mitigation strategies, we highlight the bioremediation processes, i.e., by bioaugmentation of both soil Cd-tolerant bacteria (CdtB) and cocoa bean endophytic bacteria (ECdtB), these might remove Cd in several steps of the cacao system [10]. Over the past two years, more research has been focusing on reducing high Cd content in both soils and crop management but also in the postharvest state by using several approaches without decreasing the quality of the chocolate. Hence, further research should focus on: (i) covering cacao-growing areas where no data has been recorded, either in this national survey or in other studies; (ii) include more predictive factors from several approaches, such as the geology, the biology, and the weather of the cacao systems, and (iii) assess the collected data to perform site-specific remediation packages that address the specific needs of each area and district where higher or medium Cd content in the soil has been found related to the Cd content of beans. Soil remediation methods should be addressed more widely in Colombia using this baseline of Cd in soil content to pursue the analysis of Cd in beans content in cacao-growing farms in the country.

As the aim of this manuscript was to undertake the first nationwide survey of Cd in cacao soils, this is the focus of the discussion. However, we have a partial set of data for dried cacao beans for ten out of 19 cacao-growing districts [64], In a future study, the database of Cd in cocoa beans will be expanded so that it covers the extent of soil samples analyzed in this paper. Thus, the relationship between both parameters will be performed on a large scale.

## 5. Conclusions

This study is the first in Colombia to compile a nationwide data set to: (i) determine Cd distribution of hot and coldspot clusters and their relationship with other chemical soil properties in cacao growing soils, (ii) explore the capacity of eight soil variables to predict Cd soil concentration and (iii) infer the association pattern of these soil variables. Thus, our results and recommendations set a baseline for further studies about Cd distribution and its interaction with other soil components in Colombia; moreover, it can be thought of as the first step towards the study of Cd dynamics between the soil and cacao beans in the country.

The Cd content in the soil observed nationwide was highly variable, especially in Santander. Precisely because of the high variability found in each district in the territory, it is recommended that sampling points be analyzed on the basis of their Cd content rather than by district. Hence, the agroecological zones and their corresponding weather, landscape, and geology become crucial. For instance, the samples from Santander showed the highest Cd concentrations, however, medium and lower levels of Cd were also found, which agrees with previous studies and, therefore, they are suitable for new research strategies to mitigate Cd soil content. A district with clear hotspots seems to be useful for providing concrete advice on agricultural practices and drive a mixture of remediation strategies concentrating on specific farm conditions, while a region with a fairly uniform distribution of high and low levels may require segregated harvesting, and constant monitoring. This highlights the importance of increasing the frequency of soil analysis in all cacao-growing farms and in novel scenarios where expanding the crop and mitigating the issue of Cd in both soils and beans become a reality.

**Supplementary Materials:** The following are available online at https://www.mdpi.com/article/10.3390/agronomy11040761/s1. Table S1: Summary of the predictive analysis. Table S2: Estimated degrees of the variables from the inferred partial correlation network. Table S3: Individual predictive performance summary for variables in set 8P. Table S4: Inferred partial correlation network. Figure S1: Correlogram with eight soil factors. The higher correlations were observed between Zn/Cd and Ca/Cd.

**Author Contributions:** D.B. contributed to conceptualization, methodology, formal analysis, writing the original draft and editing subsequent versions, visualization, supervision, project administration, and funding acquisition; C.L.-M. contributed to the methodology, data collection, and analysis; C.A.M. and R.V. contributed to statistical analysis and the discussion; V.M.V.-R. and G.A.A.-C. contributed to the areas of database compilation/correction, the geographical distribution of Cd in the soil, distribution of hotspots and coldspots, and the final discussion; R.Q.-M., A.Z. and E.A.G.R. contributed by collecting data and materials, participating in the discussion, and preparing the draft. All authors have read and agreed to the published version of the manuscript.

**Funding:** This research was funded by AGROSAVIA, through the grant number 1000664 for the Project entitled: Cadmium in cacao and the strategies to tackle it. The APC was funded also by Agrosavia.

**Acknowledgments:** The authors would like to thank the Colombian Ministry of Agriculture and Rural Development (MADR), the Corporación Colombiana de Investigación Agropecuaria (AGROSAVIA), and the Federación Nacional de Cacaoteros through the Fondo Nacional del Cacao (FEDECACAO) for supporting this study. The soil samples were collected according to Colombian Resolution No. 1466 of 3 December 2014, by which Agrosavia has permission to collect biological diversity samples (including soil samples) for non-commercial and scientific research purposes. To the farmers of the assessed farms who signed and agreed to soil and vegetal material sampling for research purposes, thank you. A cooperation agreement between AGROSAVIA and FEDECACAO allowed them to share their databases of soil parameters including Cd content in the soil from cacao-growing farmers analyzed in this study. The agreement was entitled: 'Convenio de cooperación técnica TV19-07 (before TV18-07)' signed by both institutions. We also would like to thank Rachel Atkinson for proofreading this manuscript and to the anonymous reviewers for their kind suggestions to improve this manuscript.

**Conflicts of Interest:** The authors declare no conflict of interest either between the institutions implied in data acquisition or in data analysis.

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
