# Peer review of "The First National Survey of Cadmium in Cacao Farm Soil in Colombia"

_agronomy, doi:10.3390/agronomy11040761_

Round 1

Reviewer 1 Report

I studied the responses. The authors give satisfactory answers to all comments. I think the paper should be accepted in this journal.

Reviewer 2 Report

The manuscript has been improved after the first review.

Reviewer 3 Report

The manuscript has been improved. I recommend to accept it in the present form.

This manuscript is a resubmission of an earlier submission. The following is a list of the peer review reports and author responses from that submission.

Round 1

Reviewer 1 Report

The manuscript with the title “The first national survey of cadmium in cacao farm soil in Colombia“ describes the spatial distribution of Cd in cacao-growing districts in Colombia and the relationship between Cd and soil parameters. The authors found that Cd was correlated with Zn and the Cd content in the soil was highly variable. I think that the manuscript could have some practical application in agriculture however there are significant flaws concerning the reason why the authors have done this work and the scientific justification of the results.

Comments

L11 Please, check the syntaxis of the sentence and the grammar

L21 “with different uses” does it concern the soils? Which could be these uses besides the agricultural one?

L23 Why the presence of Cd is one of the biggest challenges? What is the problem with these soils?

L34-37 The sentences are not well understood.

L47-49 What the authors mean by “influence Cd uptake by the cacao tree”? That the trees will uptake Cd instead of Zn and Mn? What about the toxicity of Zn and Mn in trees when are in high concentrations in soil?

L56 “increase of Cd” because of what?

L58-59 Why there is a problem with the nutrition of cocoa trees by the presence of cadmium? Which is the rationale of this observation? Are there any data concerning the critical Cd concentration in cacao plants?

L73 Did the authors estimated the “soil texture”? Are there these data of soil texture?

L74 Please, delete the first “Cd”

L75-78 It seems that these lines do not match in this paragraph.

L105 Which are the qualitative soil characteristics?

L125 I think authors should be more explanatory, what does the “pseudo-total content” represent?

L128 was instead of “were”

L139 “the standard method” which is this method?

L207-208 The high and low values deal with the Cd concentration. Which are these values of Cd that could be characterized as high and low in soil and that could be toxic and not toxic for the cacao trees?

L301 Biological neural network is not referred to as an analysis in Materials and methods.

L361 I think that the authors mean Table 1.

L363-363 Where are these data with the values of Cd? Does the ± represent the standard error or standard deviation? if so, why standard error/deviation is higher than the mean value?

L436 What is meant by the phrase “Zn had the best performance”?

L460-471 What Is the possible scientific explanation for the large correlation of Cd with Zn and Ca?

L475-476 I do not understand how this sentence is connected with the previous.

L493-494 For what reason, which is the stimulus that convinced the authors to process this research? Why did the authors measure Cd and no other heavy metal? It is not clear in the manuscript the reason that the authors have done this work.

L497 “to decrease the translocation of Cd into cacao tissue” but I think that there are no such data in this research that could support this conclusion. Is it just speculation?

Tables and Figures should be self-explanatory and explicit. For instance, in Table 1, which is the concentration unit? What does the letters n, Sd, Cv, Sk, Kt represent? In Table 3 what does the “Number” represent?

Reviewer 2 Report

This work analyses a large database related to the presence of Cd in cacao farm soils in Colombia. An elevate number of samples have been prepared and analysed for this purpose. The presentation of the data graphically using maps is nice. Moreover, the treatment of the data using predective tools enriches the discussion. In my opinion, this article can be accepted in this journal.

Reviewer 3 Report

The manuscript (MS) “The first national survey of cadmium in cacao farm soil in Colombia”, contributes to monitoring and analysis of cadmium content in cacao farm soils. The topic fits the aims and scope of the Agronomy.

It is useful research. I acknowledge the titanic effort performed by the authors to sampling and analize large data. But in my point of view, MS needs in some transformation. Here is a list of corrections to be made to the text.

  1. The abstract is too short. Please, add the details.
  2. Lines 374-375. Why the Cd contents in Colombia soils were compared with the threshold value established for agricultural soils by the Ministry of the Environment of Finland? 
  3. p.2.3. Chemical analysis. What certified reference soil materials CRMs) were used? How were recoveries of a soil CRM and detection limits?
  4. What software was used for cold/hotspots analysing?
  5. The main drawback of the MS is that it is not shown how the detected concentrations of Cd in soils affect the Cd content in cocoa fruits and other parts of the plant. It is possible that such concentrations are not deposited in cocoa, but another option can be  that soils' Cd is dangerous  even in small concentrations.
  6. Little attention has been gave to the analysis of the sources of Cd in soils. What is the Cd source in river?